# Ischemic Mitral Regurgitation: A Multifaceted Syndrome with Evolving Therapies

**DOI:** 10.3390/biomedicines9050447

**Published:** 2021-04-21

**Authors:** Mattia Vinciguerra, Francesco Grigioni, Silvia Romiti, Giovanni Benfari, David Rose, Cristiano Spadaccio, Sara Cimino, Antonio De Bellis, Ernesto Greco

**Affiliations:** 1Department of Clinical, Internal Medicine, Anesthesiology and Cardiovascular Sciences, Sapienza University of Rome, 00161 Rome, Italy; silvia.romiti_sr@libero.it (S.R.); sara.cimino@uniroma1.it (S.C.); ernesto.greco@uniroma1.it (E.G.); 2Unit of Cardiovascular Sciences, Department of Medicine Campus Bio-Medico, University of Rome, 00128 Rome, Italy; f.grigioni@unicampus.it; 3Division of Cardiology, Department of Medicine, University of Verona, 37219 Verona, Italy; benfari.giovanni@mayo.edu; 4Department of Cardiovascular Medicine, Mayo Clinic, Rochester, MN 55905, USA; 5Lancashire Cardiac Centre, Blackpool Victoria Hospital, Blackpool FY3 8NP, UK; mr.rose@nhs.net (D.R.); cristianospadaccio@gmail.com (C.S.); 6Institute of Cardiovascular and Medical Sciences, University of Glasgow, Glasgow G12 8QQ, UK; 7Department of Cardiology and Cardiac Surgery, Casa di Cura “S. Michele”, 81024 Maddaloni, Caserta, Italy; antoniodebellis@alice.it

**Keywords:** ischemic mitral regurgitation, symmetric tethering, asymmetric tethering, mitral valve repair, Mitra-Clip

## Abstract

Dysfunction of the left ventricle (LV) with impaired contractility following chronic ischemia or acute myocardial infarction (AMI) is the main cause of ischemic mitral regurgitation (IMR), leading to moderate and moderate-to-severe mitral regurgitation (MR). The site of AMI exerts a specific influence determining different patterns of adverse LV remodeling. In general, inferior-posterior AMI is more frequently associated with regional structural changes than the anterolateral one, which is associated with global adverse LV remodeling, ultimately leading to different phenotypes of IMR. In this narrative review, starting from the aforementioned categorization, we proceed to describe current knowledge regarding surgical approaches in the management of IMR.

## 1. Introduction

The right coaptation of mitral valve (MV) leaflets is achieved thanks to the balance between closing forces generated by contraction of the left ventricle (LV) and tethering forces of the subvalvular apparatus preventing leaflet prolapse into the atrium [1].

Secondary mitral regurgitation (SMR) is a consequence of geometrical modification of the mitral valve (MV) apparatus without leaflet abnormalities.

Dilated cardiomyopathy (DCM), regardless of its etiology, often leads to SMR, due to the changes in LV shape [2]. According to the general classification, the presence of coronary artery disease (CAD) affecting LV geometry and function, allows differentiation between ischemic mitral regurgitation (IMR) and functional mitral regurgitation (FMR) [3].

Impaired LV contractility due to chronic ischemia or acute myocardial infarction (AMI), often in the context of heart failure with reduced ejection fraction (HFrEF), leads to moderate and moderate-to-severe mitral regurgitation (MR) in 50% and 10% of patients, respectively [2,4,5].

IMR is mainly caused by: (1) the reduction of systolic closing forces because of the impaired LV function, (2) the displacement of papillary muscles (PM) caused by LV dilation, which increases MV leaflet tethering forces, widely recognized as the predominant causative mechanism [1].

The site of AMI exerts a specific influence determining different patterns of adverse LV remodeling. In general, inferior-posterior AMI is more frequently associated with regional structural changes than the anterolateral one, which is associated with global adverse LV remodeling [1,6,7].

As proposed by Packer et al. [8], global LV dilatation laterally displacing both PMs, in the absence of mechanical dyssynchrony, leads to SMR, as it causes symmetric tethering of mitral leaflets. Proportionately, the severity of MR follows progressive LV and annular dilatation, further increased by MR in a vicious circle feeding itself.

Therefore, the natural history of this phenotype of IMR shares a strong association with LV dysfunction and hemodynamic factors, being a predictor of adverse outcomes.

Vice versa, a disproportionate degree of MR, compared to LV dysfunction, characterizes IMR when PMs undergo asymmetrical impairment. In general, asymmetric leaflet tethering is determined by the functional involvement of the posteromedial PM as in regional LV remodeling occurring after inferior-posterior AMI. Although the chordae from the posteromedial PM are anchored to both leaflets, tethering of the posterior leaflet results in a worse distortion of mitral valve geometry.

Alternatively, in a globally dilated LV, delayed activation of anterolateral PM (mechanical dyssynchrony) as occurs in the left bundle branch block (LBBB) may represent the leading cause of asymmetric tethering [8,9].

This classification may have a crucial role in the correct management of IMR and we have graphically simplified it in the Figure 1.

Therefore, a detailed diagnostic assessment is of paramount importance. Imaging characteristics of IMR phenotypes may be a useful guide to take into account, to systematically consider the different management options.

Echocardiography is the most widely used investigation to diagnose MR, allowing the description of the etiologic and functional mechanism and the assessment of the severity of valvular regurgitation.

According to the recent European guidelines, established criteria to diagnose severe SMR are based on the echocardiographic measures of Effective Regurgitant Orifice Area (EROA) and regurgitant volume (RVol), but with lower cut-off values if compared with those for primary or organic MR; in particular EROA of >20 mm^2^ and RVol of >30 mL are enough to define severe MR. US guidelines consider higher values of both EROA and RVol for MR severity [10,11,12].

The integration of multiple techniques and parameters, including three-dimensional (3D) echocardiography assessment, is necessary to perform a comprehensive analysis [8].

MR deriving from asymmetric tethering of mitral leaflets is characterized by an eccentric and posteriorly directed jet of regurgitation; a peculiar feature of “hockey stick”, obtained by the apical and posterior movement of the tip of the posterior mitral leaflet. The “pseudo-prolapse” of the body of the anterior one is emblematic too.

On the contrary, the involvement of PMs by global dilatation of LV in the absence of mechanical dyssynchrony leads to their equal displacement and to symmetric tethering of both mitral leaflets. Qualitatively, the coaptation line is displaced apically and the resultant regurgitant jet is central [13,14,15,16,17].

Besides phenotypical categorization and magnitude of LV impairment, tethering of MV leaflets is the invariable factor characterizing IMR, making it crucial for the quantitative assessment of displacement measures to predict the severity of MR, as illustrated in Figure 2.

This narrative review was designed with the aim of comprehensively categorizing the invasive methods in the management of IMR. In particular, tailored surgical techniques addressing the aforementioned tethering phenotypes are described in the following paragraphs. 

## 2. Management of Ischemic Mitral Regurgitation

PMs dysfunction leading to MR was first described by Philips et al. [18], providing a better understanding of SMR which had previously been considered a consequence of annular dilatation and surgically approached with isolated annuloplasty [19,20].

A growing pathophysiologic complexity followed the categorization proposed by Miller et al. [21], who demonstrated the concomitant involvement of the surrounding ventricular wall.

IMR reflects different grades of LV dilatation associated with distortion of MV geometrical configuration, which potentially involves the whole apparatus.

The best surgical approach, finalized to restore MV geometry, still remains under debate.

## 3. Surgical Options

### 3.1. Mitral Valve Repair

The classification proposed by Carpentier (Table 1) has provided a paramount contribution to standardize surgical management of MR, offering a reliable link between type of MR and mitral valve repair (MV repair techniques), outlining the concept of “functional approach” [22].

Restriction of MV leaflet mobility during systole and mitral annulus dilatation, without detectable primary lesion to the integrity of MV leaflets and subvalvular apparatus, are the functional alterations found in IMR, therefore categorized as type IIIb and type I.

Consequently, for many years, mitral valve annuloplasty (MVA) has been the cornerstone of MVRepair, although in absence of a complete understanding of the pathophysiology of IMR and well-established surgical principles.

The goal of this section is to offer a systematic understanding of the most widely used MVRepair techniques in the treatment of IMR, focusing on recent evidence. Excluded from the discussion are surgical techniques at the ventricular level, external ventricular re-shapers and others, that even if they have achieved good results, are limited by a lack of experience. In particular, two different extracardiac devices were initially introduced achieving satisfactory outcomes. The CorCap (Acorn Cardiovascular, St Paul, MN, USA) cardiac support device, which has demonstrated in the Acorn trial to potentially restore LV shape when associated to restrictive mitral valve annuloplasty (rMVA) with sustained effects after 3-year and 5-year follow-ups [23,24,25].

The Coapsys (Myocor Inc., Maple Grove, MN, USA), an extracardiac device with the aim to reduce the septolateral mitral dimension and reshaping LV, has demonstrated to be effective, although in a small sample size, to improve EF and reduce MR grade [26].

### 3.2. Mitral Valve Annuloplasty

Different types of MVA techniques and annuloplasty rings were initially used in order to restore the morphology of native mitral annulus (MA) and to prevent further annular dilatation, similarly to primary MR.

The major findings of studies evaluating the effectiveness of MVA are reported in Table 2.

In patients with IMR who underwent coronary artery bypass graft (CABG), the effectiveness of adjuvant MVA to reduce the grade of regurgitation was reported, although showing an absolute unsatisfactory postoperative survival with early mortality and 5-year survival on average of 13% and 50%, respectively [37,38,39,40].

Bolling et al. [41] first introduced the surgical principle of rMVA, implanting an undersized flexible remodeling ring. The concept of this technique had the purpose to restore the competence of the MV with a reduction of the septo-lateral diameter and definitively the closure of the whole mitral orifice by the anterior mitral leaflet (AML).

Postoperative survival was surprisingly higher than former reports which adopted non-downsized MVA, with early mortality and 2-year survival being 2.1% and 72%, respectively.

The concept of rMVA was later largely adopted by other surgeons, showing good results in terms of early mortality and 5-year survival, reaching values of approximately 5% and 70%, respectively [42,43,44]. Results of a multicenter study published by Acker et al. [23], the Acorn trial, showed a reduction of early mortality (1.6%) and a similar 5-year survival, in a series of patients treated with rMVA in 84.2% of cases [25].

Although rMVA -downsizing the mitral annulus by two sizes than the one indicated by the inter-trigonal distance- has been successfully introduced among the tailored surgical strategies for treatment of IMR, discordant results concerning postoperative survival made a correct stratification of eligible patients necessary [28,45].

Interestingly, Braun et al. [28], albeit globally obtaining results in trend with former reports, stratified patients according to left ventricular end-diastolic diameter (LVEDD), highlighting relevant worse 5-year survival (49 ± 11%) in those who had LVEDD > 65 mm, synonymous of the advanced stage DCM.

Di Gianmarco et al. [27] found that preoperative functionality and volumetry of LV along with MV coaptation depth may be predictive variables for recurrence of MR after rMVA.

The authors further reported results of etiological subgroups analysis of patients with secondary MR, confirming worse long-term outcomes in the group with IMR compared to those with FMR due to idiopathic DCM.

Lacking strict selection criteria, persistence or recurrence of MR after rMVA have been reported with high frequency, representing a consistent limitation, leading to poor survival, even though the repair technique is highly reproducible and relatively easy [46,47].

The progressive and irreversible dilatation characterizing the advanced stage of DCM was recognized as the leading mechanism underlying technical failure. Nevertheless, the worse outcomes associated with the management of ischemic DCM reflected the need to carefully analyze leaflet configurations after surgical annuloplasty.

Zhu et al. [48] concluded that if the anterior mitral leaflet (AML) is not enough to cover the whole orifice due to the prevalent tethering of posterior mitral leaflet (PML), postoperative persistence of MR may be commonly found. 

Distortion of mitral valve geometry gained a leading role when Magne et al. [30] demonstrated the accuracy of preoperative posterior leaflet (PL) angle to predict persistence and recurrence of MR soon after rMVA.

Patients with PL angle ≥45 degrees, a common echocardiographic finding in the asymmetric tethering of leaflets, showed poor 3-year event-free survival and increased risk of an adverse cardiac event. 

Instead, in absence of residual MR soon after surgery, more accentuated tethering of AML was identified as the independent predictor of MR recurrence at follow-up. Gelsomino et al. [29] demonstrated that coaptation was not just a function of the AML length only, finding AL angle ≥39.5 degrees and AL excursion angle ≤35 degrees associated with increased incidence of MR recurrence.

These observations pointed out the paradoxical increase in MV geometry distortion due to asymmetric subvalvular apparatus tethering and the decisive role of meticulous echocardiographic assessment.

Additionally, when global LV dilatation is observed, a pathophysiological substrate underlying symmetric tethering, in addition to PL angle, tenting area (TA) ≥2.5 mm^2^ and tenting height (TH) ≥ 1.1 cm have been reported as valid predictive variables of technical failure [30,43].

In these patients, rMVA and the underlying surgical principle of overreduction, moving the posterior leaflet anteriorly with implantation of a symmetric and flat annuloplasty ring, was not sufficient to ensure proper coaptation of the leaflets during systole [29,30,48].

A new remodeling annuloplasty ring was designed with the purpose to act directly on the asymmetric deformation of the MV apparatus that often characterizes type IIIb ischemic MR [22]. It was named CMA IMR ETIlogix (Edwards Lifescience, Irvine, CA, USA), from the initials of its creators, Carpentier–McCarthy–Adams.

The introduction of an etiology-specific annuloplasty ring was a big step forward in the standardization of rMVA as a tailored surgical technique for IMR, which until that point was performed without an appropriate indication on the type of ring to be used.

Targeting the aforementioned asymmetric restriction of mitral leaflets, CMA IMR ETIlogix differs from the classical Physio-Ring prosthesis (Carpentier-Edwards Physio Annuloplasty Ring; Baxter-Edwards Laboratories, Irvine, CA, USA) which is flat and symmetrical, in the reduction of the posteromedial axis and configuration with smaller and depressed P2-P3 scallops [49].

The new annuloplasty ring showed excellent results significantly reducing mitral annulus diameters (MAD), TA and TH, proving to be effective in annular remodeling, in reducing leaflet tenting and in restoring leaflets coaptation.

The modifications produced on mitral valve geometry demonstrated a paramount impact in reducing early postoperative persistence and recurrence at follow-up of MR in patients with IIIb ischemic MR who underwent MV repair with the implantation of CMA IMR ETIlogix [31,32,34,36].

In addition to this remodeling annuloplasty prosthesis, the GeoForm ring (Edwards Lifesciences, Irvine, CA, USA) has been specifically designed for the surgical treatment of secondary MR; in particular, targeting the phenotype with symmetrical restriction of both mitral leaflets, caused by ischemic or idiopathic DCM.

The three-dimensional configuration of the GeoForm ring is characterized by notably reduced septo-lateral diameter, which does not make it suitable for prevalent tethering of PML, and by a posterior indentation designed to enhance leaflet coaptation and counteract subvalvular remodeling.

Votta et al. [50], by using finite element modeling, carried out an analysis on the performance of the GeoForm ring, reporting encouraging results in terms of valve competence and leaflet stress distributions, greater than those achieved with standard annuloplasty rings.

Indeed, in comparison, the Physio ring showed to require consistent undersizing to induce coaptation, causing distortion of the inter-trigonal tract, potentially altering the aortic valve mechanism and being less effective to reduce leaflet stress, mostly in severe MR.

Satisfactory results were later shown by De Bonis et al. [33], who reported their experience with the use of the GeoForm® ring (Edwards Lifesciences, Irvine, CA, USA) in the surgical treatment of secondary MR, in a series of 74 patients.

Though at 3.5 years, overall freedom from MR ≥ 2+ was 75.1 ± 8.6%, excellent outcomes were achieved considering MR with symmetric pattern of tethering, unlike predominant restriction of PML, showing a rate of persistent/recurrent MR ≥ 2+ of 9% rather than 33%, respectively.

In addition, the implantation of GeoForm® ring (Edwards Lifesciences, Irvine, CA, USA) was not associated with clinically relevant mitral stenosis, in spite of a remarkable reduction of the septo-lateral diameter [33].

Instead, after surgical annuloplasty performed with the Physio ring (Carpentier-Edwards Physio Annuloplasty Ring; Baxter-Edwards Laboratories, Irvine, CA), it was common to detect mitral stenosis (MS), labeled as “functional” for the absence of structural abnormalities of mitral leaflets, generally attributed to excessive reduction of annular size. Interestingly, the degree of functional MS was significantly related to -besides NYHA functional class- recurrent MR, emphasizing the role of postoperative subvalvular tethering in globally influencing the functionality of mitral leaflets [51].

The encouraging outcomes reported were later confirmed by Timek et al. [35], who published five-year outcomes of the implantation of GeoForm® ring (Edwards Lifesciences, Irvine, CA, USA) annuloplasty ring in patients with secondary MR.

Although the heterogeneity of patient population with almost 30% of MR grade 2+ and the great percentage of missing echocardiographic follow-up data has to be taken into account, freedom from MR ≥ 2+ at 5 years was 86%; further showing a significant impact on LV reverse remodeling with a significant increase in ejection fraction (EF) and decrease in LVEDD and LVESD, respectively.

In a recent meta-analysis, Micali et al. [52] have investigated the potential role of the type of annuloplasty ring in the recurrence of MR and LV reverse remodeling (LVRR) in patients with MR secondary to ischemic injury.

Although the study failed to identify a significant difference between ring type, rigidity/flexibility and symmetry, the best performance has been achieved by implantation of IMR-ETIlogix (Edwards Lifescience, Irvine, California, USA) which showed the lowest rate of recurrent MR (6%), followed by GeoForm® ring (Edwards Lifesciences, Irvine, CA, USA) (12%), after a mean follow-up period of 3.3 years.

The superiority of these etiology-specific annuloplasty rings may be ascribed to the ring design, which targets the different pattern of valve leaflet tethering, allowing inducing tailored changes in the geometry of MA effective to restore proper leaflet coaptation.

However, the meta-analysis, considering LVEDD at follow-up as an index for ventricular remodeling due to widely missing data about LV volumes, showed no significant LVRR after rMVA, confirming the ineffectiveness in countering LV remodeling reported by other authors [52,53,54].

### 3.3. Surgery at Sub-Valvular Level

Over the years, alongside annuloplasty, different techniques for the surgical correction of IMR to restore mitral valve continence have been proposed.

In particular, papillary muscle intervention (PMI), which has been met with greater consensus, and chordal cutting (CC) have been introduced with the purpose of counteracting the excessive tension on chordae produced by displacement of PMs [55].

In achieving this, Kron et al. [56] first described a technique for PMs relocation, which foresaw fixing the body of the posteromedial PM to the MA at the level of the right trigone. This technique was later revised by Langer and Schafers [57], approaching the subvalvular apparatus through a horizontal aortotomy. They proposed fixing the posteromedial PM to the fibrosa (midseptal annular saddle horn) underneath the commissure between noncoronary and left coronary aortic cusps under direct vision and exteriorized through the aortic wall using 3/0 pledgeted Gore-Tex. More recently, the sub-annular technique of PM repositioning has been modified by Girdauskas et al. [58], in favor, among others, to extend its use in the endoscopic mini-thoracotomy setting [59].

The concept of reducing the excessive tethering on mitral leaflets was differently approached by Hvass et al. [60], who proposed the papillary muscles approximation (PMA) technique, which consists of encircling the base of both PMs with an intraventricular sling, in order to allow repositioning towards the central line.

Initially, sharing the same surgical principle and ensuring good results when performed in addition to rMVA, surgeon preference and experience were the major determinants of choice between the two aforementioned techniques [61].

Equivalence in terms of outcomes was subsequently reported by Furukawa et al. [62], showing, among others, no differences in the incidence of MR recurrence at 3-year follow-up, although in a numerically small and heterogeneous for pathogenetic mechanisms underlying SMR study population.

The impact of PMIs in IMR was investigated in a recent meta-analysis published by Micali et al. [63], comparing outcomes achieved by the combined surgical approach, which includes PMI and MVA, versus MVA alone. The recurrence rate of MR and LVRR, defined as a reduction ≥ of 10% in LVEDD from its preoperative value, have been reported at a mean follow-up of 36.3 months.

The combined surgical approach showed a significant lower incidence of MR recurrence, highlighting the importance of correcting the primary pathophysiologic mechanism underlying MR.

In particular, PMs repositioning technique has been demonstrated to be more efficient than PMA to counteract subvalvular apparatus distortion and to allow more normal leaflet valve mobility [64,65].

These findings may be in contrast with the report published by Furukawa et al. [62], as explained by the authors in the text, it may be ascribable to the small percentage of patients with IMR in their population.

IMR is more frequently associated with asymmetric distortion of subvalvular apparatus, thus using the PMA technique, having as purpose the realignment of PMs symmetrically toward LV mid-portion, is often not appropriate to counteract the direction of tethering forces. Therefore, the use of the PMA technique should only be adopted after an accurate echocardiographic assessment of the pathophysiologic mechanism underlying FMR, identifying correctly the geometry of both the valve apparatus and LV [66].

Conversely, the meta-analysis failed to show a significant impact on LV geometry, with a reduction of LVEDD at follow-up, though slightly higher in the PMI + MVA group, still below the cut-off considered for LVRR.

Therefore, what emerged is the superiority of a surgical approach at the subvalvular level, which further allows decreasing leaflet stress distribution avoiding the use of an excessive downsized annuloplasty ring.

Fattouch et al. [67] showed excellent outcomes in terms of recurrent MR (2.7%) and LVRR at 5-year follow-up in the group that underwent PMs relocation + nonrestrictive MVA. It should be emphasized that in the inclusion criteria, severe IMR was defined by EROA ≥ 20 mm^2^ and Rvol ≥ 30 mL.

Unlike the growing evidence supporting PMI as the adjunctive technique for the management of IMR, the benefit carried by chordal cutting (CC) is yet questionable, requiring an expansion of the limited current knowledge.

The surgical principle underlying second-order CC is to increase leaflets valve mobility eliminating excessive forces of tethering.

Although clinical reports, showing good results in terms of MR recurrence and LV function, have drawn a positive consensus, controversies are mainly related to the key role played by second-order chords in connecting and ensuring stability to the main components of the MV apparatus [68,69,70].

Interestingly, Calafiore et al. [71] demonstrated that in selected patients, second-order CC may improve surgical outcomes. They specifically recruited patients with restriction of AL mobility, quantified at trans-thoracic echocardiogram (TTE) as bending angle (BA) < 145 degrees; achieving better long-term outcomes in the group underwent CC + rMVA in terms of MR recurrence and LV function, in agreement with the observations of Gelsomino et al. [29].

As hypothesized by the same authors, it seems that second-order chords have a different role from that observed in a normal setting, leading to excessive tethering and increased leaflet peak stress, paradoxically worsening LV function.

In conclusion, surgery at the subvalvular level may confer long-term durability to MVRepair, potentially reversing LV adverse remodeling. The strength of this surgical approach may be identified in the possibility of treating every single IMR phenotype in a tailored manner; the weakness lies instead in the difficulty to make the techniques easily reproducible.

The major findings of afore-described studies are reported in Table 3.

### 3.4. Surgical Mitral Plasticity

While emphasizing the efficacy of the surgical approach that combines MVA and surgery at the subvalvular level, intervention based on the concept of “mitral plasticity” should be briefly described.

This latter has been recently introduced with the purpose to describe the mechanisms adopted by the MV apparatus to compensate mechanical tethering caused by chronic ischemic LV remodeling.

Indeed, displacement of PMs triggers a series of molecular and structural changes in the MV, leading to increased length of valve leaflets, mainly the AML, and chordae tendineae, in order to allow proper coaptation. Interestingly, this adaptive process has manifested to have a variable expression with subjective clinical characteristics, showing significant IMR when it fails to provide valve competence [72,73,74].

Surgical mitral plasticity consists in the approach that plans to complete these modifications when not adequate to balance LV remodeling, including rMVA, CC and AML augmentation, has been proposed by Calafiore et al. [75].

Although surgical techniques based on leaflet augmentation, using the pericardial patch, had already been used in the past, mid-term outcomes obtained were unsatisfactory [76,77,78]. In contrast, the combined surgical approach performed by Calafiore et al. [72] showed no recurrence of 2+ MR at 1-year follow-up in patients with excessive tethering of either or both leaflets.

A significant benefit to valve leaflet mobility through the addition of second-order CC able to restore normal leaflet curvature, besides improvement of the LV function, has been demonstrated by post-operative echocardiographic findings [75].

Although these results bode well, studies with a larger sample and longer follow-up are needed in order to introduce surgery at the valvular level in the tailored management of IMR.

### 3.5. Repair vs. Replacement

The impact of correction at subannular level on LV function and MR recurrence, as recently evidenced, and its reduced use may help to better understand the discordant results in the study of MVRepair when compared with mitral valve replacement (MVR).

Cardiothoracic Surgical Trials Network (CTSN) randomized studies, designed to evaluate LVRR as the primary endpoint, compared rMVA versus MVR with preservation of the subvalvular apparatus at 2-year follow-up. They fail to observe differences in terms of LVRR and survival rate between these two surgical strategies, showing an increased number of cardiovascular readmissions in the MVRepair group (*p* = 0.01) [79,80].

However, an unequal distribution of concomitant CABG procedures among subgroups object of analysis (rMVA vs. MVR) has been reported as potential bias in the correct interpretation of data [4].

Until then, although a series of unpowered studies were reported in the initial experience, there was a consensus towards MVRepair and particularly in performing the high-reproducible technique of rMVA [40,81,82].

In contrast, a higher rate of MR recurrence, reflecting the absence of tailored techniques on valvular anatomy, as discussed in the previous section, worsened outcomes in terms of LV function and long-term survival.

Magne et al. [83] reported an almost four-fold rate of MR persistence in patients who underwent MVRepair than MVR with chordal preservation Interestingly, they performed in a limited percentage of sub-valvular surgery, including second-order CC among others, nevertheless probably not large enough in having an impact on outcomes.

Different types of annuloplasty rings were implanted in the Italian Study on the Treatment of Ischemic Mitral Regurgitation (ISTMIR), including ETIlogix and GeoForm® ring (Edwards Lifesciences, Irvine, CA, USA), specifically projected for SMR. No differences between MVRepair and MVR were found in early and late survival at 8-year follow-up and potential effects on LV performance [84].

Despite surgical principles and technical aspects, stratification of eligible patients was not-well established leading to inhomogeneous distribution among groups.

In this context, Chan et al. [85], studied outcomes of repair vs. replacement in a series of 130 patients, reporting similar long-term survival and a higher recurrence of MR in the group that underwent MVRepair. The inclusion of type II MR with leaflet prolapse, probably due to acute ischemic events, makes the methodology used unclear. Additionally, an unequal grade of LV dysfunction, more severe in MVRepair, characterized the analysis of groups.

Therefore, methodological studies, enlarging the analysis to include current surgical strategies in MVRepair and through detailed characterization of MV apparatus geometry among others, are necessary to improve our knowledge concerning the surgical approach of IMR.

Nevertheless, the choice of MVR rather than MVRepair, which more often allows a tailored restoration of MV function, specifically counteracting changes in geometry, is still an open question.

Interestingly, De Bonis et al. [86] demonstrated the superiority of MVRepair, performed with MVA and edge-to-edge technique as a bail-out, in early and late survival at a mean follow-up of 1.6 years. In particular, the great contribution of this study concerns the methodology followed in the choice of MVR, which included predictive echocardiographic parameters of technical failure such as:Extreme tethering of PML angle (>45°);Complex multiple regurgitant jets;Excessive bileaflet tethering (Ta > 4 cm^2^, CD >18 mm);Absence or mild dilatation of MA.

Excessive distortion of valvular anatomy affecting long-term durability of repair techniques may be, in suitable patients, an indication of MVR.

In this context, MVR with subvalvular apparatus preservation (SAP) may be a viable option. Indeed, it leads to a notable reduction of low-cardiac output syndrome (LCO) incidence due to the maintenance of valvular-ventricular apparatus which modulates distension and wall tension of LV during the cardiac cycle [87].

Findings from major studies reported are summarized in Table 4.

### 3.6. Moderate IMR in CABG Patients

The multifaceted complexity of IMR not only requires an appropriate choice between treatment options but also correct management timing.

As exposed in the introduction, moderate IMR complicated up to 50% of cases of patients affected by myocardial infarction (MI), definitely leading to the adverse outcomes and poor prognosis [2,10].

In HF patients with reduced ejection fraction (HfrEF), an increase in the severity of SMR was reported as predictive factor leading progressively to adverse events [88].

Though moderate SMR (EROA= 20 mm^2^; Rvol = 30 mL according to US guidelines), whose etiology mostly varies from ischemic injury to HF, exacerbates primary conditions, over the years, evidence failed to show any advantages in its adjunctive surgical management concomitant to CABG (Table 5).

Despite the more durable correction of MR, CTSN trial demonstrated the absence of a net clinical benefit in 150 CABG patients underwent rMVA due to moderate IMR.

Adjunctive MVRepair not significantly improved readmission for HF, survival and any overall adverse events; conversely, showing an early hazard of increased neurologic events and supraventricular arrhythmias, as result of more complicated surgery with longer cross-clamp and bypass times [89,90].

The conclusions drawn from their findings were that while getting better valve competence with the addition of MV repair, the absence of impact on LV function, potentially leading to reverse remodeling, translates in similar rate of outcomes [89,90]

Similar long-term outcomes may further suggest to us an equivalence in the impact of surgical approach on LV and MV apparatus, requiring an investigation on the rationale of surgical techniques in the management of moderate IMR.

Tolis et al. [91] found in the beneficial remodeling following surgical revascularization, performed alone, in patients affected by ischemic DCM and depressed EF with moderate IMR, improving regional wall motion abnormalities and PMs function, the principle at the basis of similar long-term outcomes. These observations were confirmed by studies demonstrating the efficacy of CABG performed alone to improve both IMR and functional status in the short-term [92,93,94,95].

Nevertheless, not surprisingly, some authors similarly showed that almost 40% of patients who underwent CABG alone, continued to experience at least 2+ MR [96,97]. Lam et al. [98] observed, in this kind of patient, a significant reduction of 5-year survival.

Fattouch et al. [99] reported in patients who underwent CABG with a pre-operative diagnosis of moderate IMR, a reduced 5-year survival in patients with persistence or progression of MR, with a survival rate of 73.7 ± 2.1%. In addition, among patients with depressed EF (≤ 40%), moderate IMR increased the incidence of late cardiac-related deaths.

Campwala et al. [100] studying patients with 2+ IMR, highlighted the progression of MR after CABG in 25% of cases, interestingly found independent predictors. Indeed, MR progression correlated with LV dysfunction and large LV size; incomplete revascularization of PDA area, although not showing an association with the extension of CAD and the presence of interventricular asynchrony with LBBB.

The identified predicting factors appear similar to those described in the persistence/recurrence of MR after MVRepair techniques.

Territory supplied by PDA corresponds to the inferior-posterior myocardium, frequently associated with IMR characterized by asymmetric involvement of PMs and ultimately to restriction more pronounced of PML [101].

In addition, LBBB may be similarly associated with asymmetric tethering due to the loss of contractile synchrony of PMs [9].

Therefore, we think that asymmetric tethering with distortion of MV geometry and symmetric restriction of both leaflets associated with excessive dilatation of LV, may be recognized definitively as independent predictors of MR progression when CABG alone is performed.

Ultimately, we can affirm that CABG seems to be a surgical procedure that alone cannot assure notable improvements in MV competence; thus warranting the necessity of concomitant MV surgery.

The open question remains instead why the combined approaches that included rMVA technique have failed to improve outcomes.

As we have seen in the paragraph on MVRepair, in order to ensure excellent results, long-term durability of MV repair should be unavoidably obtained through an accurate and detailed study of valvular anatomy, further identifying potential predictors of technical failure. In this context, rMVA is addressed to restore valve competence reducing MA diameters, having no role in counteracting sub-valvular tethering forces and definitively LV function [102].

In fact, when rMVA is tailored to MR etiology, using a specific annuloplasty ring may be associated with good short-term outcomes, as demonstrated by Chan et al. [103], who implanted ETIlogix® in the 85% of patients in the combined MVRepair plus CABG group. They reported a significant improvement at 1-year follow-up of functional capacity, LVRR, MR severity and B-type natriuretic peptide levels (BNP), compared with CABG alone.

However, long-term outcomes may be modified only by surgical techniques addressing specifically geometrical changes of MV apparatus.

Therefore, randomized trials are necessary to improve our knowledge in order to highlight the role of concomitant MV sub-valvular repair techniques as crucial options in the management of moderate IMR.

Major findings from prospective studies described are reported in Table 5.

## 4. Percutaneous Options

### 4.1. Mitra-Clip

The surgical principle behind MitraClip system (Abbott Vascular Devices, Santa Clara, CA, USA), in order to improve MV competence, consists of tissue approximation leading to the increased surface of coaptation, recognizing its forerunner in the surgical suture-based approach [104].

This latter has been introduced by Alfieri in 1991 and labeled as edge-to-edge surgical technique, initially targeting cases of severe MR in which, due to peculiarity of the lesion, the traditional techniques of reconstruction were challenging [105].

MitraClip is based on a transcatheter approach, including a clip system delivered into the left atrium through a transeptal puncture performed in the fossa ovalis.

Three-dimensional trans-esophageal echocardiogram (TEE) represents the essential tool both to guide towards a detailed assessment of MV anatomy and to target the lesion (Figure 3).

Reduction of MR may be achieved by correct positioning of the clips between free edges of leaflets and the subsequent release.

The feasibility of the procedure with a percutaneous approach and the reversibility of implantable clips, with the intraoperative possibility of repositioning them until obtaining the satisfactory results, have favored the large diffusion of MitraClip [105].

The EVEREST II trial [106] was designed to study the effectiveness and safety of transcatheter mitral leaflet approximation with MitraClip device compared with MV surgery in the management of severe MR. The results showed that conventional surgery was more effective to reduce MR grade than percutaneous repair. In the MitraClip group, 23% of patients showed persistence of MR ≥ 2+ and further at 12 months the rate of surgery required due to MV dysfunction was significantly higher.

In contrast, percutaneous repair has been demonstrated to be superior in safety at 30 days, with the rate of major adverse events 3-fold higher in the surgery arm and to achieve similar clinical outcome improvements [106]. Nevertheless, as highlighted by George et al. [107], symptoms and LV size may be influenced by the higher recurrence of MR ≥ 2+ in affecting long-term outcomes.

Though controversies have accompanied EVEREST II trial findings, important considerations have been drawn from subgroups analysis, based on the pathophysiology of MR, which have sharply marked the MitraClip future application.

Indeed, in the small subgroup of patients with secondary MR, both percutaneous and surgical MV repair were associated with similar outcomes.

Therefore, the need for an accurate stratification of eligible patients began to be clear.

Baldus et al. [108] reported results from the German transcatheter mitral valve interventions (TRAMI) registry; 481 patients underwent percutaneous edge-to-edge therapy using MitraClip, of which 93% had severe MR classified in the two-thirds of cases of functional nature.

Patients were selected by the Heart Team mainly for the prohibitive risk for the surgical approach, showing low in-hospital mortality (2.5%) but considerable post-discharge mortality (12.5%) with a median follow-up of 3 months, reflecting the advanced stage of the disease.

Similarly, Maisano et al. [109] enrolled patients with a surgical high-risk profile, elderly and who affected by FMR, confirming good outcomes in terms of hospital mortality and major adverse events, further improving quality of life (QoL) and functional status at 1-year follow-up.

Technical feasibility, procedural safety and clinical efficacy by using MitraClip have been the benefit reported more recently by Geis et al. [110], who demonstrated a significant reduction of MR in the 98.8% of patients affected by severely impaired LV function.

Results from two randomized trials MITRA-FR (Percutaneous Repair with the MitraClip Device for Severe Functional/Secondary Mitral Regurgitation) and COAPT (Cardiovascular Outcomes Assessment of the MitraClip Percutaneous Therapy for Heart Failure Patients with Functional Mitral Regurgitation) have been published [111,112].

The trials have been designed to compare MitraClip in addition to optimal medical therapy (OMT) (intervention group) and OMT alone (control group) in the management of patients with systolic HF and secondary MR.

Although COAPT trial showed the effectiveness and safety analysis better in the intervention group, MITRA-FR fails to demonstrate any improvements in the prognosis of patients treated with percutaneous MV repair in comparison with the control group [111,112].

Several commentaries tried to explain the discordant results obtained by these two trials, apparently designed with an equal setting, sharing the same basic ideas.

Indeed, patient selection, percutaneous procedures and medical treatment process are recognized as the main criticisms [113,114,115].

In comparison to MITRA-FR, LV size in COAPT was smaller whereas FMR was more severe (ERO ≈ 0.41 cm^2^ vs. 0.31 cm^2^), characterizing treated cases for a disproportionate grade of MR when correlated to LV adverse remodeling, as postulated by Grayburn et al. [115].

Therefore, considering LV size, in the study COAPT, the severity of MR was more associated to volume overload, worsening prognosis while the LV is still recoverable.

In MITRA-FR, the proportionate grade of MR reflected an advanced stage of CDM and the only intervention at the valvular level failed to reverse positively a LV globally dilated, in which tethering forces not counteracted further increased residual MR at 1-year [113,114,115].

The viewpoint reported by Grayburn et al. [115], by using the correlation between LV size and MR grade, emphasize the necessity to distinguish between the two different patterns of FMR; in fact, we can easily switch the terms used by the authors with that referred to FMR pathophysiologic nature.

As afore-described in the text, IMR with symmetric tethering is more often the marker of the advanced stage of CDM and in contrast, asymmetric tethering, usually secondary to LV regional remodeling, is the result of a more pronounced distortion of MV geometry leading to disproportionate MR grade.

Moreover, some authors already demonstrated that patients presenting with more dilated LV and MV annulus, higher values of pulmonary pressures, could derive less benefit from MitraClip implant [116,117].

In conclusion, MitraClip is a viable option in patients with a surgical high-risk profiles, nevertheless, in order to correctly establish its role in the management of IMR, an integrative assessment including a detailed study of MV anatomy and patients’ characteristics is mandatory. On the other hand, no prospective studies comparing surgery for IMR versus optimal medical treatment (OMT) in the setting of significant LV dysfunction are yet available.

### 4.2. Annuloplasty

Cardioband system (Edwards Lifescience, Irvine, CA, USA) is a transcatheter-based approach mimicking the traditional annuloplasty, in which the device is directly delivered into the left atrium through the femoral vein and a transseptal puncture [118].

In contrast, the Carillon™ Mitral Contour System™ (Cardiac Dimensions™ Inc., Kirkland, WA, USA) encircles MA indirectly being delivered in the coronary sinus (CS) through access performed in the external jugular vein [119].

Similarly to MitraClip, the target population is represented by surgical high-risk profile patients with symptomatic FMR despite OMT.

Experience with these devices is still limited, although results are encouraging.

Cardioband system (Edwards Lifescience, Irvine, CA, USA) showed better clinical outcomes at 12 months follow-up when compared with the MitraClip device [120,121].

In a randomized trial, the Carillon device has demonstrated its safety and efficacy to significantly reduce MR and LV volumes in symptomatic FMR patients when added to OMT [122].

### 4.3. Transcatheter Mitral Valve Implantation (TMVI)

Differently from transcatheter aortic valve implantation (TAVI), MV replacement with trans-catheter therapy has not achieved clinical consensus due to the complexity of MV anatomy and disease pathophysiology.

Mainly for this reason, although several devices have been described in the literature, the use of TMVI is still limited to compassionate use or very high-risk patients [123].

### 4.4. Cardiac Resynchronization Therapy (CRT)

The rationale at the basis of the use of cardiac resynchronization therapy (CRT) in the management of IMR, consists in counteracting restriction of MVL, increasing closure forces [124].

Indeed, Kanzaki et al. [125] recognized dyssynchrony of PMs as the leading mechanism associated with MR in patients with HF and LBBB, rapidly improved by using CRT, which provide coordinated mechanical activation of PMs.

In confirming the acute effects of CRT to target PMs dyssynchrony, changes in echocardiographic parameters of IMR severity have been reported, reduction of ERO among others, reflecting the improved clinical status of patients [126,127,128,129].

In addition, the restoration of physiological electrical sequence has demonstrated a long-term effect, after 3–6 months of CRT, on LV dimensions and shape, leading to LVRR and increasing global contractility, further improving MV competence [130].

The direct correlation at long-term between LV remodeling and FMR improvements has been confirmed by Verhaert et al. [131], who further identified the magnitude of IMR acute improvements and the residual MR after CRT as independent predictors of outcomes at 1-year follow-up.

Similarly, Van Bommel et al. [132] reported that improvement in IMR at 6 months after CRT predicted better outcomes at 2-year follow-up. The 49% of patients who underwent CRT implantation in their study showed an improvement in IMR grade.

In contrast, predictors of technical failure are still poorly understood due to the complexity of the condition, not allowing definitively the standardization of this technique in the tailored management of IMR in HF patients with LBBB.

Dibiase et al. [133] showed that ischemic etiology was associated with fewer improvements in IMR when treated by CRT.

Sitges et al. [134] found in mitral tenting area >3.8 cm^2^ the strongest predictor of MR persistence, expression of high grade IMR. Excessive dilatation of LV was identified by Ohnishi et al. [135] together with a scar at PMs insertion site and the absence of strain dyssynchrony, as predictive factors of unsatisfactory outcomes.

Interestingly, Gavazzoni et al. [136], with the purpose to guide further studies, have proposed three different classes of tailored strategies based on probability to be a responder to CRT, considering the predictive factors aforementioned.

## 5. Conclusions

IMR, regardless of its severity, represents an independent predictor of poor outcome.

MVA has been the first widely used MVRepair technique in the surgical approach of severe IMR. Criticisms associated with technical failure and the long-term durability of this approach associated with the growing pathophysiologic knowledge, mainly related to echocardiographic developments, have been the catalytic factors that moved towards a targeted surgical approach.

Gradually, the flat and symmetrical annuloplasty ring was replaced by prosthesis which reflected the MV geometrical changes, as seen with CMA IMR ETIlogix and GeoForm annuloplasty rings, achieving better results in terms of MR recurrence at early and late follow-up after surgery.

In contrast, concomitant MVA and sub-valvular surgical correction, including the afore reported techniques in targeting MV apparatus abnormalities, may restore a more physiologic configuration.

The surgical approaches addressing PMs, although limited by still low sample size, have demonstrated superior outcomes in reducing the restriction of mitral leaflets, increasing MV competence and mainly in having an impact on LV volumes and contractility.

Therefore, a comprehensive approach to severe IMR should be addressed first to properly categorize MV structural changes, allowing to quantify geometrical distortion of valve apparatus, further evaluating echocardiographic parameters potentially predictive of technical failure. Quantitative echocardiographic measurements such as CD, MVTa, TnV, interpapillary distance among others, should be followed by an accurate assessment of tethering angles and degree of leaflets restriction.

Echocardiography should offer itself as a direct line toward the possibility of a combined surgical approach, including MVA and subvalvular surgery when feasible, as the preferred choice of management [137,138,139,140,141,142].

On the other hand, the excessive geometrical distortion of MV, in particular affecting the length and mobility of valve leaflets, or a significantly dilated LV, which increase the risk of technical failure, may be the indication to choose MVR. 

In patients requiring surgical revascularization, with moderate IMR, due to the long-term poor prognosis and the demonstrated ineffectiveness achieved by MVA alone, surgical approach to MV could be avoided.

In the modern era of the percutaneous approach, minimally invasive techniques such as MitraClip have been largely used, even though the poor understanding of the complexity of IMR.

The consistent experience achieved in performing MVA has demonstrated that surgical correction performed at the annular level is not enough to provide long-term durability of the repair.

In contrast, CRT offers a different point of view in the management of IMR, even though requiring a proper categorization of eligible patients, it needs to be encouraged in the context of a combined surgical approach.

## Figures and Tables

**Figure 1 biomedicines-09-00447-f001:**
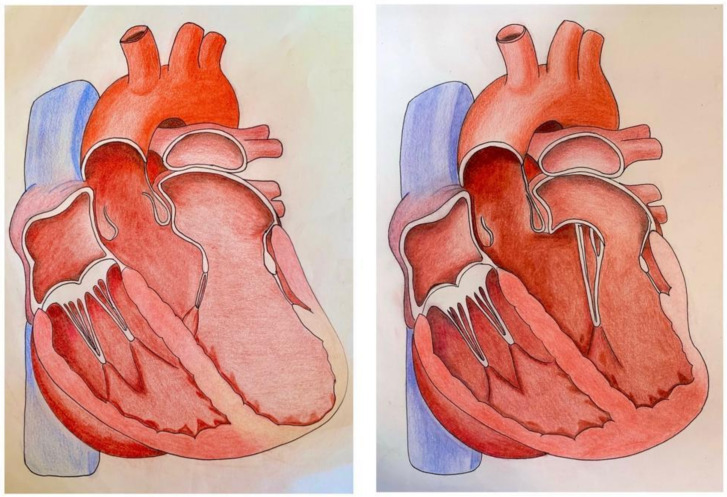
The two main different phenotypes of ischemic mitral regurgitation (IMR): on the left, left ventricle (LV) is globally dilated, displacement of papillary muscles (PMs) is symmetrical, leading to symmetric systolic tethering of mitral valve (MV) leaflets; on the right, inferior-posterior acute myocardial infarction (AMI) causes asymmetric tethering of MV leaflets with an excessive systolic restriction of posterior MV leaflet.

**Figure 2 biomedicines-09-00447-f002:**
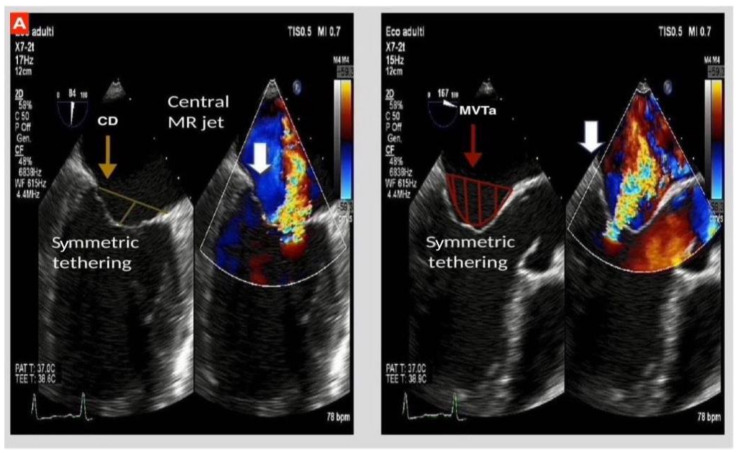
Transesophageal echocardiogram (TEE); (**A**): IMR with symmetric tethering characterized by central jet of regurgitation in a globally dilated left ventricle with increased mitral annulus diameter. On the left, coaptation depth (CD) may be identified as the distance between the coaptation and the annular plane; on the right, mitral valve tenting area (MVTa) is seen as the space confined between valve leaflets and annular plane. (**B**): IMR with asymmetric tethering of mitral valve leaflets, more accentuated for posterior mitral leaflet (PML), the white arrows highlight the eccentric jet of mitral regurgitation in the presence of a left ventricle which is not globally dilated.

**Figure 3 biomedicines-09-00447-f003:**
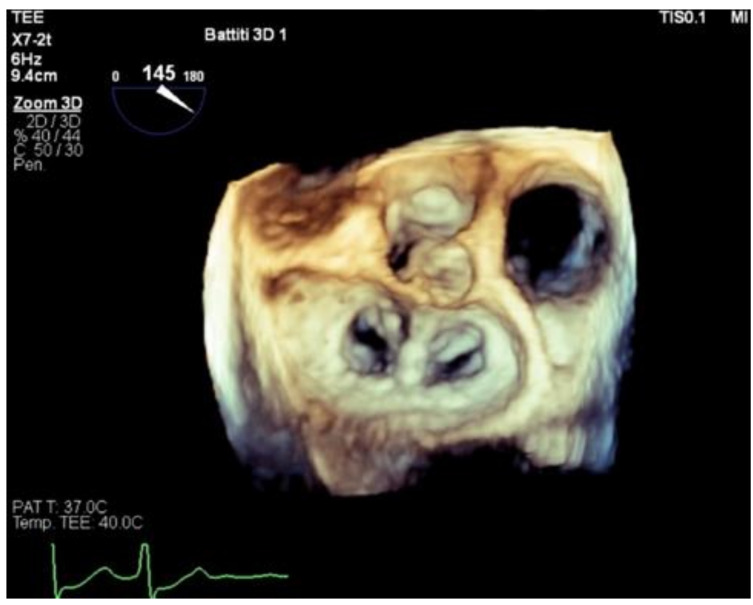
MitraClip implantation with characteristic double-orifice valve at three-dimensional (3D) trans-esophageal echocardiography (TEE).

**Table 1 biomedicines-09-00447-t001:** Carpentier surgical classification of MV pathology.

Carpentier Classification	Definition
Type I	Normal leaflet mobility
Type II	Increased mobility
Type III	Restricted mobility; during diastole (3A); during systole (3B);

**Table 2 biomedicines-09-00447-t002:** Findings from major studies evaluating the effectiveness in terms of freedom from mitral regurgitation recurrence after mitral valve annuloplasty (MVA) in the management of severe secondary mitral regurgitation (SMR), further reporting analyzed clinical outcomes.

Source	No.	Study Design	Years	Type of Ring/Downsizing	Rate of Concomitant CABG	Freedom from MR Recurrence Early * and at Late f/u	Main Findings
**Di Gianmarco et al.** [27], **2007**	142 (73.94% ischemic)	Single-center retrospective	1997–2005	DeVega-like annuloplasty (21%); Glutaraldehyde-treated autologous pericardium (45%); Miniband flexible ring (Sorin) (34%); Overreduction of MV posterior annulus	99 of 105 (94.30%)	- 65.5 ± 8.3% of MR < 2+ at 4-year f/u	Ischemic DCM is associated with poorer outcomes. The main predictive variables for MR recurrence were LVEDV, CD, LVEF in the whole group; LVESV and LVEF in the ischemic group.
**Braun et al.** [28], **2008**	100	Single-center retrospective	2000–2004	PhysioRing Downsized by two sizes	100 (100%)	- 85% of MR < 2+ at 5-year f/u	LVEDD > 65 mm as predictor of poor outcome after rMVA
**Gelsomino et al.** [29], **2008**	251	Single-center prospective	2001–2007	- Downsized by two sizes	251 (100%)	- 72.2% at 5-year f/u had MR ≥ 2	Outcome at 5-year f/u in terms of freedom from re-operation for failed repair and index for LVRR were unsatisfactory, outlining the poor long-term durability of rMVA.
**Magne et al.** [30], **2007**	51 (retrospective) 17 (prospective)	Single-center retrospective and prospective	2002–2005	PhysioRing Downsized by two sizes CMA-IMR ETIlogix (3.92%)	49 of 51 (96.08%) 17 of 17 (100%)	22% (11 of 51) of persistent MR in the retrospective series 5.88% (1 of 17) of persistent MR in the prospective series	PL angle ≥ 45 degrees is a predictive echocardiographic parameter of technical failure
**Daimon et al.** [31], **2006**	59	Multicenter prospective	2003–2005	CMA IMR ETIlogix ring	37 of 59 (62.70%)	3% of MR persistence 10 days after surgery	Reduction of MR, MAD and leaflet tethering with targeted annuloplasty ring.
**Filsoufi et al.** [32], **2007**	40	Single-center prospective	2003–2005	CMA-IMR ETIlogix	27 of 40 (68%)	No persistence of MR At median f/u of 15 months freedom from recurrent MR ≥ 2+ was 97%	Excellent durability of repair technique
**De Bonis et al.** [33], **2011**	74 (64% ischemic)	Single-center prospective	2005–2008	GeoForm ring	33 of 74 (44.60%)	5% of persistent MR ≥ 2+ At 3.5 years f/u freedom from MR ≥ 2+ was 75.1 ± 8.6	Persistent/Recurrent MR ≥ 2+ was 33% in patients with preoperative asymmetric tethering versus 9% in symmetric tethering
**Mosquera et al.** [34], **2009**	35	Single-center prospective	2005–2008	CMA IMR ETIlogix	31 of 35 (88.60%)	2.86% persistent MR ≥ 2+ At 25 months freedom from MR ≥ 2+ was 88.9%	Excellent mid-term outcomes
**Timek et al.** [35], **2014**	86	Single-center prospective	2005–2011	GeoForm ring	67 of 86 (78%)	- At 5-year f/u freedom from MR ≥ 2+ was 86%	Low recurrent MR rate at f/u
**Campisi et al.** [36], **2016**	157	Single-center prospective	2006–2012	CMA IMR ETIlogix	100 of 157 (63.70%)	No persistence of MR Freedom from MR ≥ 2+ was 96.6% at median f/u of 28 months	Excellent durability of repair technique

* early f/u: in-hospital/thirty day f/u.; the dashes (-) in the Table indicate missing data; MR: mitral regurgitation; DCM: dilatative cardiomyopathy; LVEDV: left ventricular end-diastolic volume; CD: coaptation depth; LVEF: left ventricular ejection fraction; LVESV: left ventricular end-systolic volume; NYHA: New York Heart Association; LVEDD: left ventricular end-diastolic diameter; LVRR: left ventricular reverse remodeling; rMVA: restrictive mitral valve annuloplasty; MAD: mitral annulus diameter.

**Table 3 biomedicines-09-00447-t003:** Findings from major studies evaluating the effectiveness in terms of freedom from mitral regurgitation recurrence and left ventricular reverse remodeling after adjuvant sub-valvular surgery in the management of severe IMR, further reporting analyzed clinical outcomes.

Source	No.	Study Design	Years	Type of Adjuvant Subvalvular Surgery Techique	Freedom from MR Recurrence	Main Findings
**Borger et al.** [70], **2007**	92 (46.74% CC)	Single-center prospective	1998–2005	MVA + CC vs. MVA alone	At 2-year f/u recurrent MR ≥ 2+ was 15% in the CC group versus 37% in the group MVA alone (*p* = 0.03)	Preoperatively LV function was worse in CC group and similar in both groups in the post-operative period
**Wakasa et al.** [61], **2015**	45 (100% PMA) cPMA in 71.11% iPMA in 28.89%	Single-center retrospective	1999–2013	MVA + cPMA vs. MVA + iPMA	The 4-year survival rate and rate of freedom from recurrence of MR ≥ 2+ were 83% and 85% for those underwent cPMA rather than 48% and 48% for those with iPMA.	Complete PMA was associated with lower postoperative mortality and high durability of valve repair
**Hvass et al.** [60], **2003**	10 (100% PMA)	Single-center retrospective	June 2000–	MVA + PMA	No residual MR early and late at f/u (maximum 24 months)	Reduction in LV dimensions Improved NYHA functional class
**Fattouch et al.** [67], **2014**	115 (100% PMrel)	Single-center prospective	2003	MVA + PMrel	Recurrence of MR ≥ 2+ at 5-year f/u was 2.7% Significant LVRR	Excellent results from PMrel technique Severe MR identified with EROA ≥ 0.2 cm^2^ and RVol ≥ 30 mL
**Langer et al.** [57], **2009**	60 (50% PMrel)	Single-center prospective	2004–2009	MVA + PMrel vs. MVA	Persistenst MR I-II + of 3% in both groups Freedom from recurrent MR ≥ 2+ in PMrel group was 94% versus 71% (*p* = 0.01) Significant LVRR in PMrel group (*p* < 0.001)	Better outcomes for PMrel group
**Calafiore et al.** [71], **2014**	67 (46.27% CC)	Single-center prospective	2007–2011	rMVA + CC vs. MVA	Recurrent MR less in CC group at median f/u of 33 months (*p* = 0.014) LVRR statistically significant in CC group (*p* = 0.022 and *p* = 0.029 for diastolic and systolic dimension, respectively)	Eligible patients underwent adjuvant CC were with BA < 145 angles
**Nappi et al.** [66], **2016**	96 (50% PMA)	Prospective randomized clinical trial	2007–2010	MVA + PMA vs. rMVA	LV significant improvements in PMA group (*p* < 0.001)	Long-term beneficial effects on LVRR and MV geometrical configuration even though survival rate was similar
**Furukawa et al.** [62], **2018**	18 (100% SVR)	Single-center retrospective	2010–2016	MVA + PMA ± PMrel ± CC	Recurrent rate of MR ≥ 2+ at 3-year and 5-year f/u was 97%	Long-term durability of MV sub-annular repair techniques targeted to MV abnormalities
**Harmel et al.** [64], **2019**	101 (50.50% PMrel)	Single-center prospective	2016–2018	rMVA + PMrel vs. rMVA	Recurrent MR ≥ 2+ at 1-year f/u was 98% vs. 86.7% in the PMrel group and rMVA alone group, respectively (*p* = 0.045)	Excellent outcomes for rMVA + PMrel
**Pausch et al.** [65], **2019**	108 (55.56% PMrel)	Single-center prospective	2016–2018	MVA + PMrel	No residual MR early after surgery Recurrent MR ≥ 2+ at 1-year f/u was 3.3 % in the PMrel group and 20.8% in the rMVA group, respectively (*p* = 0.037)	Excellent outcomes for adjuvant PMrel technique at 1-year f/u

MR: mitral regurgitation; MVA: mitral valve annuloplasty; CC: chordal cutting; LV: left ventricle; PMA: papillary muscle approximation; cPMA: complete papillary muscle approximation; iPMA: incomplete papillary muscle approximation; BA: bending angle; rMVA: restrictive mitral valve annuloplasty; PMrel: papillary muscle relocation; EROA: effective regurgitant orifice area; NYHA: New York Heart Association; RVol: regurgitant volume;.LVRR: left ventricular reverse remodeling; SVR: sub-valvular repair.

**Table 4 biomedicines-09-00447-t004:** Findings from major studies comparing freedom from mitral regurgitation recurrence and clinical outcomes between mitral valve replacement and repair in the management of severe IMR.

Source	No.	Study Design	Years	Repair Rate	Freedom from MR Recurrence	Outcome	Main Findings
**Magne et al.** [83], **2009**	370	Single-center retrospective	1995–2008	50%	Higher persistence of MR in MV repair group	Lower operative mortality in MV repair group (*p* = 0.03) Similar survival rate at median f/u of 3.8 years (*p* = 0.17)	MV repair is not superior to MVR in terms of operative and overall mortality
**LoRusso et al.** [84], **2013**	1006	Multi-center registry	1996–2011	70.4% (ETIlogix in 3.3%; GeoForm in 1.6%)	Freedom from recurrent MR ≥ 2+ was 75%	MV repair had lower 30 days in-hospital and late (8 years f/u) mortality, although not statistically significant (*p* = 0.32; *p* = 0.42) Freedom from all-cause reoperation and valve related reoperation higher in MVR group (*p* < 0.001)	MV repair was a strong predictor of reoperation
**De Bonis et al.** [86], **2012**	132	Single-center prospective	2000–2009	64.4% (MVA ± edge-to edge repair)	Freedom from recurrent MR ≥ 2+ was 78.3% in MV repair group with a rate of paravalvular leak of 9.7% in MVR group at 5.5 years f/u	Significant improvements of LV function and dimensions in MV repair group (*p* = 0.0001)	In patients with advanced dilated and ischemic DCM, MVR is associated with higher in-hospital and late mortality than in MV repair group
**Chan et al.** [85], **2011**	130	Single-center prospective	2001–2010	50%	Recurrent MR ≥ 2+ at late f/u was 23% in MV repair and 2% in MVR	Similar freedom from valve-related complications and similar LV function at f/u (*p* > 0.2)	MVR remains a viable option for the treatment of IMR
**Acker et al.** [79], **2014**	251	Prospective randomized clinical trial	2009–2012	50%	Recurrence of mitral regurgitation at 12 months was higher in the MVrepair group than in the MVR group (32.6% vs. 2.3%, *p* < 0.001).	LV dimensions and function Survival rate Adverse events and hospitalization Quality of life	No difference between LVRR and survival at 1 year f/u
**Goldstein et al.** [80], **2016**	251	Prospective randomized clinical trial	2009–2012	50%	Recurrence of MR higher in MVrepair group with 58.8% vs. 3.8% (paravalvular leak) in MVR group, respectively at 2 years f/u (*p* < 0.001)	LV dimensions and function (*p* = 0.18) Survival rate Adverse events and hospitalization Quality of life	No difference between LVRR and survival at 2 years f/u

MV: mitral valve; MVR: mitral valve replacement; LVRR: left ventricular reverse remodeling; MR: mitral regurgitation; DCM: dilatative cardiomyopathy; IMR: ischemic mitral regurgitation.

**Table 5 biomedicines-09-00447-t005:** Findings from major studies comparing clinical outcomes of adjuvant MVA in CABG patients versus standalone CABG in the management of moderate IMR.

Source	No.	Study Design	Repair Rate	Outcome	Main Findings
**Chan et al.** [103], **2012**	73	Multi-center single-blinded randomized controlled trial	46.58% (CMA IMR ETIlogix annuloplasty ring)	Peak oxygen consumption LVESVI MR volume Plasma B-type natriuretic peptide Deaths at 30 days and at 1-year f/u	MVA plus CABG in moderate ischemic MR may improve functional capacity (*p* < 0.001), LVRR (*p* = 0.002) and MR severity (*p* = 0.001)
**Smith et al.** [89], **2014**	301	Randomized prospective clinical trial	50%	LVRR Deaths at 30 days and at 1-year f/u Adverse events Hospitalization Quality of life	No difference between MVA plus CABG and CABG alone Reduced prevalence of moderate MR but an increased number of untoward events
**Michler et al.** [90], **2016**	301	Randomized prospective clinical trial	50%	LVRR Deaths at 2-year f/u Adverse events Hospitalization Quality of life	Durable correction of MR that failed to reflect any significant difference in terms of outcomes between the two groups

MVA: mitral valve annuloplasty; CABG: coronary artery bypass graft; MR: mitral regurgitation; LVESVI: left ventricular end-systolic volume index; LVRR: left ventricular reverse remodeling.

## Data Availability

Not applicable.

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
