# Peer review of "Ischemic Mitral Regurgitation: A Multifaceted Syndrome with Evolving Therapies"

_biomedicines, 2021, doi:10.3390/biomedicines9050447_

Round 1
Reviewer 1 Report
The authors have apparently edited the manuscript very substantially again. The revised version appears to constitute an authoritative review.
Author Response
Authors fully appreciate the reviewer’s comment.
Reviewer 2 Report
The revised manuscript is significantly improved.
I think omitting the previous 1st part was the right thing to do, since now the reader can easier read through the entire manuscript
Major Comments.
- Are there any data regarding the effect of EF (reduced vs preserved) in either of the surgical interventions that you describe? You mention it in the section with CABG, but it would be more complete if you adda short paragraph separately or in the relevant sections.
Minor Comments
- Conclusions sections is improved , however , I would delete the paragraph “nevertheless…..alone”. Additionally, I feel that the paragraphs “ the surgical approaches….contractility” and “ nevertheless, surgery….conclusions” are similar , and one of the two could be deleted as well
- In central illustration I would delete the phrase “ potentially….apparatus”
- Page 2, paragraph 3. In IMR do you mean FMR?
- Figure 1, the echo B on the left is confusing since it is ischemic MR and has central regurgitant jet. Please correct the “simmetric “ that appears in the images as well
- Page 5, paragraph 6: after” ..or experience” , I would add and research data.
- Page 9, paragraph 5, where you say adverse remodeling , do you mean reverse? Please correct
- Page9, the paragraph “ these findings….techniques” doesn’t make a lot of sense, can you rephrase or delete?
- Page 10, paragraph 8, delete the word “nevertheless”
- Page 12, paragraph 1, would replace “ surgery” with intervention, . In general I would also describe what is the surgical mitral plasticity “technique”
- Page 13, paragraph 1, would replace “surgery” with correction
- Page 13, paragraph 9, correct “well-not” with not-well
- Page 15, paragraph 3: “ progressive detrimental effects” of what? Please explain/add
- Page 15, paragraph 10. The phrase “Lam et al….IMR” doesn’t make a lot of sense, please rephrase
- Page 16, paragraph 5: you can rephrase like” CABG alone seems to be a surg procedure that alone cant…..”
- Page 18, paragraph 15 please rephrase” worsening notable prognosis”
- Page 19. Percutaneous annuloplasty is only tested in limited clinical trials, therefore it is not”commonly used” please rephrase.
- Page 21, paragraph 1, replace “may “ with could
- Page 21, mitraclip is not any more limited in use. Please rephrase or delete
- Page 21, paragraph 4, replace “surgery “ with surgical correction. Also rephrase from “repair worsening….surgical patients”
Author Response
REVIEWER 2
The revised manuscript is significantly improved. I think omitting the previous 1st part was the right thing to do, since now the reader can easier read through the entire manuscript.
Major Comments.
- Are there any data regarding the effect of EF (reduced vs preserved) in either of the surgical interventions that you describe? You mention it in the section with CABG, but it would be more complete if you adda short paragraph separately or in the relevant sections.
Authors fully appreciate the reviewer’s suggestions. We have revised studies regarding surgical intervention, highlighting the lack of analysis centered on the effect of EF. Accordingly, in order to make clearer the section, we have rephrased the paragraph in which we report potential effect of EF.
We think that a further discussion about MR in HFpEF patients should expand excessively our review.
In fact, as underlined alongside the manuscript, ischemic mitral regurgitation is associated with tethering of mitral leaflets due to displacement of papillary muscles in the context of a reduced EF with dilated LV. The decrease in systolic function of LV is in fact one of the determinants of IMR severity.
In contrast, HF with preserved EF, even though often associated to SMR , recognizes a different pathophysiologic mechanism, in which the diastolic dysfunction and the onset of atrial fibrillation, lead to predominant mitral valve annular dilatation.
Minor Comments
- Conclusions sections is improved, however, I would delete the paragraph “nevertheless…..alone”. Additionally, I feel that the paragraphs “the surgical approaches….contractility” and “ nevertheless, surgery….conclusions” are similar , and one of the two could be deleted as well.
Authors are thankful to the reviewer for their comments. We have provided to delete the aforementioned paragraphs.
- In central illustration I would delete the phrase “ potentially….apparatus”.
Authors are thankful to the reviewer for their comments. We have deleted the phrase.
- Page 2, paragraph 3. In IMR do you mean FMR?
Authors are thankful to the reviewer for their comments. In the paragraph we referred to both IMR and FMR, we have provided to change with the term SMR.
- Figure 1, the echo B on the left is confusing since it is ischemic MR and has central regurgitant jet. Please correct the “simmetric “ that appears in the images as well.
Authors are thankful to the reviewer for their comments. We have improved Figure 1 correcting in the images the term symmetric and removing the figure on the left, in order to make clearer for readers to distinguish between the two different phenotypes.
- Page 5, paragraph 6: after” ..or experience” , I would add and research data.
Authors fully appreciate the reviewer’s concern. We have provided to report the more important publications in the cited field.
- Page 9, paragraph 5, where you say adverse remodeling , do you mean reverse? Please correct.
Authors are thankful to the reviewer for their comments. As suggested, we have changed accordingly.
- Page9, the paragraph “ these findings….techniques” doesn’t make a lot of sense, can you rephrase or delete?
Authors are thankful to the reviewer for their comments. We have decided to delete the paragraph.
- Page 10, paragraph 8, delete the word “nevertheless”.
Authors are thankful to the reviewer for their comments. We have deleted the word, as suggested.
- Page 12, paragraph 1, would replace “ surgery” with intervention, . In general I would also describe what is the surgical mitral plasticity “technique”.
Authors are thankful to the reviewer for their comments. We have replaced the term and introduced the concept of surgical mitral plasticity “technique”.
- Page 13, paragraph 1, would replace “surgery” with correction.
Authors are thankful to the reviewer for their comments. We have replaced the term.
- Page 13, paragraph 9, correct “well-not” with not-well.
Authors are thankful to the reviewer for their comments. We have corrected.
- Page 15, paragraph 3: “ progressive detrimental effects” of what? Please explain/add.
Authors are thankful to the reviewer for their comments. We have rephrased the paragraph.
- Page 15, paragraph 10. The phrase “Lam et al….IMR” doesn’t make a lot of sense, please rephrase.
Authors are thankful to the reviewer for their comments. We have rephrased the paragraph.
- Page 16, paragraph 5: you can rephrase like” CABG alone seems to be a surg procedure that alone cant…..”.
Authors are thankful to the reviewer for their comments. We have rephrased the paragraph, as suggested.
- Page 18, paragraph 15 please rephrase” worsening notable prognosis”.
Authors are thankful to the reviewer for their comments. We have rephrased the paragraph.
- Page 19. Percutaneous annuloplasty is only tested in limited clinical trials, therefore it is not”commonly used” please rephrase.
Authors are thankful to the reviewer for their comments. We have rephrased the paragraph.
- Page 21, paragraph 1, replace “may “ with could.
Authors are thankful to the reviewer for their comments. We have replaced, as suggested.
- Page 21, mitraclip is not any more limited in use. Please rephrase or delete.
Authors are thankful to the reviewer for their comments. We have rephrased the paragraph.
- Page 21, paragraph 4, replace “surgery “ with surgical correction. Also rephrase from “repair worsening….surgical patients”.
Authors are thankful to the reviewer for their comments. We have rephrased the paragraph.

This manuscript is a resubmission of an earlier submission. The following is a list of the peer review reports and author responses from that submission.
Round 1
Reviewer 1 Report
To authors
Vinciguerra et al, wrote a review article on the pathophysiology and the treatment of ischemic MR.
The manuscript is lengthy, which may induce, readers' fatigue, especially when reaching the last parts of the article which are, per my opinion, well written and more important than the first.
I would recommend splitting the manuscript into 2 parts, one for pathophysiology and imaging of ischemic MR(if the authors still want to write about this), and a second one about the invasive methods of treating ischemic MR.
the first part is not well written and needs major revision. The 2nd part , as already mentioned , is better written , has adequate literature search, and shows that the authors have an expertise in that area.
A general major comment would be, that, per my opinion, it would be better to use the wording primary vs secondary(functional) MR and the secondary could then be categorized into the ischemic MR vs functional MR in the absence of CAD.
This general classification, (used in ESC Guidelines, Braunwald' s valvular disease textbook) should also be explained in the first part.
Regarding the first part the authors are confusing the cardiac remodeling increased wall stress and cardiac muscle loss, in their effort to explain the systolic dysfunction. It should be explained whether ischemic MR can occur only in HF with reduced EF or even in HFpEF(with no systolic dysfunction)
They try to explain the mechanism of fibrosis, changes on a cellular level, but I would recommend describing the valvular apparatus and the macroscopic changes that are happening in ischemic MR instead.
Imaging of MR is better presented, however needs also a lot of work.
Central illustration needs explanation, the figures that are used, I don't think that they are having significant educational value for the readers (they just show different echos of MR, with no explanation, of the pathophysiology with arrows/heads etc). ie. You could compare in the same image the asymmetric vs symmetric tethering. I think a figure explaining the angles a, b, tenting area and the other quantitative parameters would be more helpful.(and readers could easier understand the difference between symmetric vs asymmetric tethering)
The surgical options part is well written, scientifically sound, and shows the authors' expertise in this area.
It needs revision in terms of language, syntactical errors and typos.
Author Response
Vinciguerra et al, wrote a review article on the pathophysiology and the treatment of ischemic MR.
The manuscript is lengthy, which may induce, readers' fatigue, especially when reaching the last parts of the article which are, per my opinion, well written and more important than the first.
I would recommend splitting the manuscript into 2 parts, one for pathophysiology and imaging of ischemic MR(if the authors still want to write about this), and a second one about the invasive methods of treating ischemic MR.
the first part is not well written and needs major revision. The 2nd part , as already mentioned , is better written , has adequate literature search, and shows that the authors have an expertise in that area.
Authors are thankful to the reviewer for their comments. The manuscript has been revised according their indications, conceptually splitting it into 2 parts, the first shorter, where we describe macroscopic changes of mitral valve apparatus from a pathophysiologic and echocardiographic point of view; while in the second one we focus on surgical management of IMR.
A general major comment would be, that, per my opinion, it would be better to use the wording primary vs secondary(functional) MR and the secondary could then be categorized into the ischemic MR vs functional MR in the absence of CAD.
This general classification, (used in ESC Guidelines, Braunwald' s valvular disease textbook) should also be explained in the first part.
Authors are thankful to the reviewer for their comments. In the manuscript we have categorized MR according to the suggested general classification of ESC guidelines, selectively using the term IMR when we refer to MR due to coronary artery disease. We also changed the title according this suggestion.
Regarding the first part the authors are confusing the cardiac remodeling increased wall stress and cardiac muscle loss, in their effort to explain the systolic dysfunction. It should be explained whether ischemic MR can occur only in HF with reduced EF or even in HFpEF(with no systolic dysfunction)
They try to explain the mechanism of fibrosis, changes on a cellular level, but I would recommend describing the valvular apparatus and the macroscopic changes that are happening in ischemic MR instead.
Authors fully appreciate the reviewer’s concern. We have removed the section regarding changes in LV remodeling on a cellular level, after acute myocardial infarction, focusing more on the structural changes involving the MV apparatus.
Imaging of MR is better presented, however needs also a lot of work.
Central illustration needs explanation, the figures that are used, I don't think that they are having significant educational value for the readers (they just show different echos of MR, with no explanation, of the pathophysiology with arrows/heads etc). ie. You could compare in the same image the asymmetric vs symmetric tethering. I think a figure explaining the angles a, b, tenting area and the other quantitative parameters would be more helpful. (and readers could easier understand the difference between symmetric vs asymmetric tethering).
Authors are thankful to the reviewer for their comments. We have added a comprehensive explanation to the central illustration and reorganized the echocardiographic figures; the new Figure 1 allows to compare the two phenotypes of tethering, the asymmetric vs symmetric, further showing the rationale at the basis of measurements of the main quantitative parameters by echo.
The surgical options part is well written, scientifically sound, and shows the authors' expertise in this area.
Authors fully appreciate the reviewer’s comment.
It needs revision in terms of language, syntactical errors and typos.
Authors are thankful to the reviewer for their comments. We have thoroughly revised the language.

Reviewer 2 Report
This is very interesting review.
I do not have any comments.
Author Response
This is very interesting review. I do not have any comments.
Authors fully appreciate the reviewer’s concern.

Reviewer 3 Report
In this review on ischemic functional mitral regurgitation, the authors provide an extensive analysis of this difficult and multidimensional topic. My main criticisms are the lack of coherence and clarity in the flow of data.
1. The first sentence of the abstract does not make sense. There is a double negation: absence of abnormalities.
2. Style and language in the abstract should be improved. It would be appropriate that the whole revised paper is edited by a native English speaker before resubmission.
3. Line 39: do not capitalise acute myocardial infarction.
4. Page 2: every one sentence or two sentences appear to be a new paragraph in the PDF. That is peculiar. A paragraph should contain a logical flow of several concepts that constitute one coherent entity.
5. Line 85. It is not clear what the authors mean by pressure overload in this context. Afterload represents all the factors that contribute to total myocardial wall stress during systolic ejection. Afterload=(SP) (SR)/ 2WT where SP is systolic pressure, SR is systolic radius and WT is wall thickness. Or, the law of Laplace for a thick-walled cylinder states that the wall stress is the pressure multiplied by the radius and divided by two times the wall thickness. According to the law of Laplace, any dilatation of the ventricle will lead to an increase of the diastolic and systolic wall stress, and thereby stimulates further enlargement of the left ventricle.
6. Once again, there is a lack of coherence of the text on pages 2 and 3. There should be a logical flow of concepts.
7. The number of non-standard abbreviations is way too high.
8. The therapeutic dimension of the paper starts at line 291. I am not opposed to a solid introduction and I agree the order of the different sections is correct. However, in the management section, there is again the problem that the authors present a series of data or statements but without a clear coherence or a clear message. At the end of the day, I want to know how therapeutic decisions are taken based on which criteria and potentially on which specific algorithm. How solid is the foundation of such a decision making process? This is certainly not as easy as the interpretation of randomised trials because many decisions are taken on rational grounds rather than on formal empirical proof. Formal proof is very difficult to obtain in this setting.
9. Again, the number of paragraphs on different pages of the management section appears to be countless. Coherence of data, ideas, and concepts are needed.
10. The conclusion brings more clarity. Even when there is no consensus in the literature, it would be good to know which algorithm the authors follow based on personal experience and based the literature. A good synthesis with all parameters taking into account in the decision making process would be helpful.
Author Response
In this review on ischemic functional mitral regurgitation, the authors provide an extensive analysis of this difficult and multidimensional topic. My main criticisms are the lack of coherence and clarity in the flow of data.
- The first sentence of the abstract does not make sense. There is a double negation: absence of abnormalities.
Authors are thankful to the reviewer for their comments. We have modified the mentioned sentence.
- Style and language in the abstract should be improved. It would be appropriate that the whole revised paper is edited by a native English speaker before resubmission.
Authors are thankful to the reviewer for their comments. We have largely revised the language.
- Line 39: do not capitalise acute myocardial infarction.
Authors are thankful to the reviewer for the comments. We have corrected the aforementioned phrase.
- Page 2: every one sentence or two sentences appear to be a new paragraph in the PDF. That is peculiar. A paragraph should contain a logical flow of several concepts that constitute one coherent entity.
Authors fully appreciate the reviewer’s comment. We have reorganized the ‘‘pathophysiology’’ paragraph, removing the discussion regarding LV remodeling and focusing more on macroscopic changes of the MV apparatus, in order to achieve more clarity and coherence.
- Line 85. It is not clear what the authors mean by pressure overload in this context. Afterload represents all the factors that contribute to total myocardial wall stress during systolic ejection. Afterload=(SP) (SR)/ 2WT where SP is systolic pressure, SR is systolic radius and WT is wall thickness. Or, the law of Laplace for a thick-walled cylinder states that the wall stress is the pressure multiplied by the radius and divided by two times the wall thickness. According to the law of Laplace, any dilatation of the ventricle will lead to an increase of the diastolic and systolic wall stress, and thereby stimulates further enlargement of the left ventricle.
Authors are thankful to the reviewer for their comments. We have decided to remove the pathophysiological description of LV remodeling.
- Once again, there is a lack of coherence of the text on pages 2 and 3. There should be a logical flow of concepts.
Authors are thankful to the reviewer for their comments. We have simplified concepts expressed into the ‘‘pathophysiology’’ paragraph centering the narrative on the MV.
- The number of non-standard abbreviations is way too high.
Authors are thankful to the reviewer for their comments. We have standardized abbreviations.
- The therapeutic dimension of the paper starts at line 291. I am not opposed to a solid introduction and I agree the order of the different sections is correct. However, in the management section, there is again the problem that the authors present a series of data or statements but without a clear coherence or a clear message. At the end of the day, I want to know how therapeutic decisions are taken based on which criteria and potentially on which specific algorithm. How solid is the foundation of such a decision making process? This is certainly not as easy as the interpretation of randomized trials because many decisions are taken on rational grounds rather than on formal empirical proof. Formal proof is very difficult to obtain in this setting.
Authors fully appreciate the reviewer’s concern. We have read the management section in light of their comment and we have simplified and made more coherent sub-paragraphs, further adding conclusive lines in order to express a clear take-home message for every management strategy discussed.
- Again, the number of paragraphs on different pages of the management section appears to be countless. Coherence of data, ideas, and concepts are needed.
Authors are thankful to the reviewer for their comments. The management section has been reviewed to be more coherent, linking the narrative between the different sub-paragraphs.
- The conclusion brings more clarity. Even when there is no consensus in the literature, it would be good to know which algorithm the authors follow based on personal experience and based the literature. A good synthesis with all parameters taking into account in the decision making process would be helpful.
Authors are thankful to the reviewer for their comments. The conclusion has been enriched, discussing the crucial role of echocardiography in the decision making process. However, it is difficult to identify an accurate algorithm followed in the potential preferred choice of management due to the still poorly understood complexity of the disease.

Round 2
Reviewer 1 Report
The revised manuscript is improved.
The annotated figures are now more helpful, and authors did a very good job.
However, it is still lengthy (29 pages!), and requires a lot of effort from the reader to start and finish it at one time.
I think that the authors have realized this and haven’t included important analytical data, while discussing the invasive techniques for this reason.
Authors seem to have an expertise in this topic, however, several sections of their discussion are description of data without connection to one another.
Several sections of the manuscript need to be rechecked and revised regarding the language and style.
Additionally, it lacks a clear message, which should be stated in the conclusions section.
Major Comments.
- I still believe that the authors should omit the 1st part and focus on describing the various invasive methods.
- It should be noted that ischemic regurgitation is not only the acute AMI, but also the chronic with ischemia only one, since in the first part they use the former condition (not wrong, just need to be defined)
- Regarding the mitral valve annuloplasty it would be good to mention whether there was concomitant CABG performed and what were the outcomes.
- It would be helpful to add the Carpentier’s MR classification as a table for reader’s convenience
- Conclusion should be more concise and have a better message from repair vs MVR +- CABG
Minor Comments
- The manuscript should be rechecked for language and style
- In figures 1,2 it should be noted that they are TEEs, and in figure 2 correct the word “symmetric”
- In Paragraphs in page 14 “Zhu et al….orifice”, page 20 “napi et al ..analysis” , it is not clear what the authors want to underline and need to be revised.
- In page 23, line 734, do you mean severe or moderate MR per ESC guidelines? If so, correct accordingly even in table 4.
- Some of the data from important studies, described in discussion, should be more detailed, ie in page 21, lines 681-685, what were the actual numbers of early (what was early, in-hospital, 1 month?) and 8 year fup mortality with p values. At least those numbers should be analytical in the respective tables
Reviewer 3 Report
The revised paper is clearly an improvement but the starting point was very weak. One reviewer rejected the paper, one did not really comment, and I gave a very diplomatic assessment. We need at least one additional round of revision, in particular to improve style and language of the paper.